# Dynamic Local Regret for Non-convex Online Forecasting

**Sergul Aydore** *
Department of ECE
Stevens Institute of Technology
Hoboken, NJ USA
sergulaydore@gmail.com

**Tianhao Zhu**
Department of ECE
Stevens Institute of Technology
Hoboken, NJ USA
romeo.zhuth@gmail.com

**Dean Foster**
Amazon
New York, NY USA
foster@amazon.com

## Abstract

We consider online forecasting problems for non-convex machine learning models. Forecasting introduces several challenges such as (i) frequent updates are necessary to deal with concept drift issues since the dynamics of the environment change over time, and (ii) the state of the art models are non-convex models. We address these challenges with a novel regret framework. Standard regret measures commonly do not consider both dynamic environment and non-convex models. We introduce a local regret for non-convex models in a dynamic environment. We present an update rule incurring a cost, according to our proposed local regret, which is sublinear in time $T$. Our update uses time-smoothed gradients. Using a real-world dataset we show that our time-smoothed approach yields several benefits when compared with state-of-the-art competitors: results are more stable against new data; training is more robust to hyperparameter selection; and our approach is more computationally efficient than the alternatives.

## 1 Introduction

Our goal is to design efficient stochastic gradient descent (SGD) algorithms for training non-convex models for online time-series forecasting problems. A time series is a temporally ordered sequence of real-valued data. Time series applications appear in a variety of domains such as speech processing, financial market analysis, inventory planning, prediction of weather, earthquake forecasting; and many other similar areas. Forecasting is the task of predicting future outcomes based on previous observations. However, in some domains such as inventory planning or financial market analysis, the underlying relationship between inputs and outputs change over time. This phenomenon is called *concept drift* in machine learning (ML) [Žliobaitė et al., 2016]. Using a model that assumes a static relationship can result in poor accuracy in forecasts. In order to address concept drift, the model should either be periodically re-trained or updated as new data is observed.

Recently, the state of the art in forecasting has been dominated by models with many parameters such as deep neural networks [Flunkert et al., 2017, Rangapuram et al., 2018, Toubeau et al., 2019]. In large scale ML, re-training such complex models using the entire dataset will be time consuming. Ideally, we should update our model using new data instead of re-training from scratch at every time step. Offline (batch/mini-batch) learning refers to training an ML model over the entire training dataset. Online learning, on the other hand, refers to updating an ML model on each new example as it is observed. Using online learning approaches, we can make our ML models deal with concept drift efficiently when re-training over the entire data set is infeasible.

The performance of online learning algorithms is commonly evaluated by regret, which is defined as the difference between the real cumulative loss and the minimum cumulative loss across number of updates [Zinkevich, 2003]. If the regret grows linearly with the number of updates, it can be concluded that the model is not learning. If, on the other hand, the regret grows sub-linearly, the model is learning and its accuracy is improving. While this definition of regret makes sense for convex optimization problems, it is not appropriate for non-convex problems, due to NP-hardness of non-convex global optimization even in offline settings. Indeed, most research on non-convex problems focuses on finding local optima. In literature on non-convex optimization algorithms, it is common to use the magnitude of the gradient to analyze convergence [Hazan et al., 2017, Hsu et al., 2012]. Our proposed dynamic local regret adopts this framework, defining regret as a sliding average of gradients.

Standard regret minimization algorithms efficiently learn a static optimal strategy, as mentioned in [Hazan and Seshadri, 2009]. But this may not be optimal for online forecasting problems where the environment changes due to concept drift. To cope up with dynamic environments, some have proposed efficient algorithms for adaptive regrets [Daniely et al., 2015, Zhang et al., 2018, Wang et al., 2018]. However, these works are limited to convex problems. Our proposed regret extends the dynamic environment framework to non-convex models.

**Related Work:** Online forecasting is an active area of research [Kuznetsov and Mohri, 2016]. There is a rich literature on linear models for online forecasting [Anava et al., 2013, Koolen et al., 2015, Liu et al., 2016, Hazan et al., 2018, Gultekin and Paisley, 2019]. Among these, Kuznetsov and Mohri [2016], Koolen et al. [2015], Hazan et al. [2018] study the online forecasting problem in the regret framework. The regret considered in [Hazan et al., 2018] adapts to the dynamics of the sytem but it is limited to linear applications.

The most relevant work to our present contribution is Hazan et al. [2017], which introduced a notion of *local regret* for online non-convex problems. They also proposed efficient algorithms that have non-linear convergence rate according to their proposed regret. The main idea is averaging the gradients of the most recent loss functions within a window that are evaluated at the current parameter. However, such regret definition of local regret assumes a static best model. This paper precisely addresses both non-convexity and dynamic environment for online forecasting problems in a novel regret framework.

**Our Contributions:** We present a regret framework for training non-convex forecasting models in dynamic environments. Our contributions:

- We introduce a novel local regret and demonstrate analytically that it has certain useful properties, such as robustness to new training data.
- We present an update rule for our regret: a dynamic exponentially time-smoothed SGD update. We prove that it is sublinear according to our proposed regret.
- We show that on a benchmark dataset our approach yields stability against new data and robustness to hyperparameter selection.
- Our approach is more computationally efficient than the algorithm proposed by Hazan et al. [2017]. We show this empirically on a benchmark dataset, and discuss why it is the case.

We provide extensive experiments using a real-world data set to support our claims. All of our results can be reproduced using the code in `https://github.com/Timbasa/Dynamic_Local_Regret_for_Non-convex_Online_Forecasting_NeurIPS2019`.

## 2   Setting

In online forecasting, our goal is to update the model parameters $x_t$ at each time step $t$ in order to incorporate the most recently available information. Assume that $t \in \mathcal{T} = \{1, \cdots, T\}$ represents a collection of $T$ consecutive points where $T$ is an integer and $t = 1$ represents an initial forecast point. $f_1, \cdots, f_T : \mathcal{K} \to \mathbb{R}$ are non-convex loss functions on some convex subset $\mathcal{K} \subseteq \mathbb{R}^d$. Then $f_t(x_t)$ represents the loss function computed using the data from time $t$ and predictions from the model parameterized by $x_t$, which has been updated on data up to time $t - 1$. In the subsequent sections, we will assume $\mathcal{K} = \mathbb{R}^d$.

## 2.1 Static Local Regret

Hazan et al. [2017] introduced a local regret measure based on gradients of the loss. Using gradients allows the authors to address otherwise intractable non-convex models. Their regret is local in the sense that it averages a sliding window of gradients. Their regret quantifies the objective of predicting points with small gradients on average. They are motivated by a game-theoretic perspective, where an adversary reveals observations from an unknown static loss. The gradients of the loss functions from the $w$ most recent rounds of play are evaluated at the current model parameters $x_t$, and these gradients are then averaged. More formally, Hazan et al. [2017]'s local regret, we call Static Local Regret, is defined to be the sum of the squared magnitudes of the average gradients as in Definition 2.1.

**Definition 2.1.** (Static Local Regret) The $w$-local regret of an online algorithm is defined as:

$$SLR_w(T) \triangleq \sum_{t=1}^{T} \|\nabla F_{t,w}(x_t)\|^2 \tag{1}$$

when $\mathcal{K} = \mathbb{R}^d$ and $F_{t,w}(x_t) \triangleq \frac{1}{w} \sum_{i=0}^{w-1} f_{t-i}(x_t)$. Hazan et al. [2017] proposed various gradient descent algorithms where the regret $SLR$ is sublinear.

The motivation behind averaging is two-fold: (i) a randomly selected update has a small time-averaged gradient in expectation if an algorithm incurs local regret sublinear in $T$ (ii) for any online algorithm, an adversarial sequence of loss functions can force the local regret incurred to scale with $T$ as $\Omega(\frac{T}{w^2})$. These arguments presented in Hazan et al. [2017] inspire our use of local regret. However, static local regret computes loss from the past $f_{t-i}$ using the most recent parameter $x_t$. In other words, the model is evaluated on in-sample data. This can cause problems for forecasting applications because of concept drift. For instance, consider a demand forecasting problem where your past loss $f_{t-i}$ represents your objective in November and $x_t$ represents the parameters of your model for January in the following year. Assuming that the sales increase in November due to Christmas shopping, evaluating November's objective using January's parameters can be misleading for decision making.

## 2.2 Proposed Dynamic Local Regret

Here, we introduce a new definition of a local regret that suits forecasting problems motivated by the concept of *calibration* [Foster and Vohra, 1998] . First we consider the first order Taylor series expansion of the cumulative loss. The loss function calculated using the data at time $t$ is $f_t$. The model parameters trained on data up to $t - 1$ are $x_t$. We perturb $x_t$ by $u$:

$$\sum_{t=1}^{T} f_t(x_t + u) \approx \sum_{t=1}^{T} f_t(x_t) + \sum_{t=1}^{T} \langle u, \nabla f_t(x_t) \rangle \tag{2}$$

for any $u \in \mathbb{R}^d$. If the updates $\{x_1, \cdots, x_T\}$ are *well-calibrated*, then perturbing $x_t$ by any $u$ cannot substantially reduce the cumulative loss. Hence, it can be said that the sequence $\{x_1, \cdots, x_T\}$ is asymptotically calibrated with respect to $\{f_1, \cdots, f_T\}$ if: $\limsup_{T \to \infty} \sup_{\|u\|=1} \frac{\sum_{t=1}^{T} f_t(x_t) - \sum_{t=1}^{T} f_t(x_t + \delta u)}{T} \leq 0$ where $\delta$ is a small positive scalar. Consequently, using the first order Taylor series expansion, we can write the following equation that motivates the left hand side of equation 3: $\limsup_{T \to \infty} \sup_{\|u\|=1} -\frac{1}{T} \langle u, \nabla f_t(x_t) \rangle \leq 0$. Hence, by controling the term $\sum_{t=1}^{T} \langle u, \nabla f_t(x_t) \rangle$, we ensure asymptotic calibration. In addition, we can write the following lemma for the upper bound of this first order term as:

**Lemma 2.2.** For all $x_s$, the following equality holds:

$$\sup_{\|u\|=1} \sum_{s=t-w+1}^{t} \langle u, \nabla f_s(x_s) \rangle = \left\| \sum_{s=t-w+1}^{t} \nabla f_s(x_s) \right\|. \tag{3}$$

Based on the above observation, we propose the regret in Definition 2.3. The idea is exponential averaging of the gradients at their corresponding parameters over a window at each update iteration.

**Definition 2.3.** (Proposed Dynamic Local Regret) We propose a $w$-local regret as:

$$DLR_w(T) \triangleq \sum_{t=1}^{T} \left\| \frac{1}{W} \sum_{i=0}^{w-1} \alpha^i \nabla f_{t-i}(x_{t-i}) \right\|^2 = \sum_{t=1}^{T} \|\nabla S_{t,w,\alpha}(x_t)\|^2$$

where $S_{t,w,\alpha}(x_t) \triangleq \frac{1}{W}\sum_{i=0}^{w-1}\alpha^i f_{t-i}(x_{t-i})$, $W \triangleq \sum_{i=0}^{w-1}\alpha^i$, and $f_t(x_t) = 0$ for $t \leq 0$. The motivation of introducing $\alpha$ is two-fold: (i) it is reasonable to assign more weights to the most recent values, (ii) having $\alpha$ less than 1 results in sublinear convergence as introduced in Theorem 3.4.

Using our definition of regret, we effectively evaluate an online learning algorithm by computing the exponential average of losses by assigning larger weight to the recent ones at the corresponding parameters over a sliding window. We believe that our definition of regret is more applicable to forecasting problems than the static local regret as evaluating today's forecast on previous loss functions might be misleading.

**Motivation via a Toy Example** We demonstrate the motivation of our dynamic regret via a toy example where the Static Local Regret fails. Concept drift occurs when the optimal model at time $t$ may no longer be the optimal model at time $t + 1$. Let's consider an online learning problem with concept drift with $T = 3$ time periods and loss functions: $f_1(x) = (x - 1)^2, f_2(x) = (x - 2)^2, f_3(x) = (x-3)^2$. Obviously, the best possible sequence of parameters is $x_1 = 1, x_2 = 2, x_3 = 3$. We call this the *oracle policy*. Also consider a suboptimal sequence, where the model does not react quickly enough to concept drift: $x_1 = 1, x_2 = 1.5, x_3 = 2$. We call this the *stale policy*. The values of the *stale policy* were chosen to minimize Static Local Regret. Using the formulation of static and dynamic local regrets, we can write:

$$SLR_3(3) = \left\|\frac{\nabla f_3(x_3) + \nabla f_2(x_3) + \nabla f_1(x_3)}{3}\right\|^2 + \left\|\frac{\nabla f_2(x_2) + \nabla f_1(x_2)}{3}\right\|^2 + \left\|\frac{\nabla f_1(x_1)}{3}\right\|^2 \quad \text{(Hazan's)}$$

$$DLR_3(3) = \left\|\frac{\nabla f_3(x_3) + \nabla f_2(x_2) + \nabla f_1(x_1)}{3}\right\|^2 + \left\|\frac{\nabla f_2(x_2) + \nabla f_1(x_1)}{3}\right\|^2 + \left\|\frac{\nabla f_1(x_1)}{3}\right\|^2 \quad \text{(Ours)}$$

Note that, for the local regrets, we use $w = 3$ and assume $f_t(x) = 0$ for $t \leq 0$. We also set $\alpha = 1$ for our Dynamic Local Regret but other values would not change the results for this example. The formulation of the Standard Regret is $\sum_{t=1}^{T} f_t(x_t) - \min_x \sum_{t=1}^{T} f_t(x)$. Although the *oracle policy* achieves globally minimal loss, the Static Local Regret favors the *stale policy*. We can verify this by computing the loss and regret for these policies, as shown in the Table 1.

| Regret | Oracle Policy | Stale Policy | Decision |
|---|---|---|---|
| Cumulative Loss | 0 | 5/4 | Oracle policy is better |
| Standard Regret | -2 | -3/8 | Oracle policy is better |
| Static Local Regret (Hazan et al.) | 40/9 | 4/9 | Stale policy is better |
| Dynamic Local Regret (Ours) | 0 | 10/9 | Oracle policy is better |

Table 1: Values of various regrets for the toy example. The Static Local Regret incorrectly concludes that the stale policy is better than the oracle policy.

## 2.3 Dynamic Exponentially Time-Smoothed Stochastic Gradient Descent

Below we present two algorithms based on SGD algorithms which are shown to be effective for large-scale ML problems [Robbins and Monro, 1951]. Algorithm 1 proposed in [Hazan et al., 2017] represents the static time-smoothed SGD algorithm which is sublinear according to the the regret in Definition 2.1 . Here, we propose to use dynamic exponentially time-smoothed online gradient descent represented in Algorithm 2 where gradients of loss functions are calculated at their corresponding parameters. Stochastic gradients are represented by $\hat{\nabla} f$.

---
**Algorithm 1** Static Time-Smoothed Stochastic Gradient Descent (STS-SGD)
---
**Require:** window size $w \geq 1$, learning rate $\eta > 0$, Set $x_1 \in \mathbb{R}^n$ arbitrarily
  1: **for** $t = 1, \cdots, T$ **do**
  2:    Predict $x_t$. Observe the cost function $f_t : \mathbb{R}^b \to \mathbb{R}$
  3:    Update $x_{t+1} = x_t - \frac{\eta}{w}\sum_{i=0}^{w-1}\hat{\nabla} f_{t-i}(x_t)$.
  4: **end for**
---

**Algorithm 2** Dynamic Exponentially Time-Smoothed Stochastic Gradient Descent (DTS-SGD)

---

**Require:** window size $w \geq 1$, learning rate $\eta > 0$, exponential smoothing parameter $\alpha \to 1^-$ (means that $\alpha$ approaches 1 from the left), normalization parameter $W \triangleq \sum_{i=0}^{w-1} \alpha^i$, Set $x_1 \in \mathbb{R}^n$ arbitrarily
1: **for** $t = 1, \cdots, T$ **do**
2:     Predict $x_t$. Observe the cost function $f_t : \mathbb{R}^b \to \mathbb{R}$
3:     Update $x_{t+1} = x_t - \frac{\eta}{W} \sum_{i=0}^{w-1} \alpha^i \hat{\nabla} f_{t-i}(x_{t-i})$.
4: **end for**

---

Note that STS-SGD needs to perform $w$ gradient calculations at each time step, while we perform only one and average the past $w$. This is a computational bottleneck for STS-SGD that we observed in our experimental results as well.

## 3 Main Theoretical Results

In this section, we mathematically study the convergence properties of Algorithm 2 according to our proposed local regret. First, we assume the following assumptions hold for each loss function $f_t$: (i) $f_t$ is bounded: $| f_t(x) | \leq M$ (ii) $f_t$ is L-Lipschitz: $| f_t(x) - f_t(y) | \leq L\|x - y\|$ (iii) $f_t$ is $\beta$-smooth: $\|\nabla f_t(x) - \nabla f_t(y)\| \leq \beta\|x - y\|$ (iv) Each estimate of the gradient in SGD is an i.i.d random vector such that: $\mathbb{E}\left[\hat{\nabla} f(x)\right] = \nabla f(x)$ and $\mathbb{E}\left[\|\hat{\nabla} f(x) - \nabla f(x)\|^2\right] \leq \sigma^2$. Using the update in Algorithm 2, we can define the update rule as: $x_{t+1} = x_t - \eta \tilde{\nabla} S_{t,w,\alpha}(x_t)$. Note that each $\tilde{\nabla} S_{t,w,\alpha}(x_t)$ is a weighted average of $w$ independently sampled unbiased gradient estimates with a bounded variance $\sigma^2$. Consequently, we have:

$$\mathbb{E}\left[\tilde{\nabla} S_{t,w,\alpha}(x_t) \mid x_t\right] = \nabla S_{t,w,\alpha}(x_t)$$

$$\mathbb{E}\left[\|\tilde{\nabla} S_{t,w,\alpha}(x_t)) - \nabla S_{t,w,\alpha}(x_t)\|^2 \mid x_t\right] \leq \frac{\sigma^2(1 - \alpha^{2w})}{W^2(1 - \alpha^2)}. \tag{4}$$

As a result of the above construction, we have the following lemma for the upper bound of $\|\nabla S_{t,w,\alpha}(x_t)\|^2$.

**Lemma 3.1.** For any $\eta$, $\beta$, $\alpha$ and $w$, the following inequality holds:

$$\left(\eta - \frac{\beta}{2}\eta^2\right) \|\nabla S_{t,w,\alpha}(x_t)\|^2 \leq S_{t,w,\alpha}(x_t) - S_{t+1,w,\alpha}(x_{t+1})$$

$$+ S_{t+1,w,\alpha}(x_{t+1}) - S_{t,w,\alpha}(x_{t+1}) + \eta^2 \frac{\beta}{2} \frac{\sigma^2(1 - \alpha^{2w})}{W^2(1 - \alpha^2)} \tag{5}$$

Next, we compute upper bounds for the terms in the right hand side of the inequality in Lemma 3.1.

**Lemma 3.2.** For any $0 < \alpha < 1$ and $w$ the following inequality holds:

$$S_{t+1,w,\alpha}(x_{t+1}) - S_{t,w,\alpha}(x_{t+1}) \leq \frac{M(1 + \alpha^{w-1})}{W} + \frac{M(1 - \alpha^{w-1})(1 + \alpha)}{W(1 - \alpha)} \tag{6}$$

**Lemma 3.3.** For any $0 < \alpha < 1$ and $w$, the following inequality holds:

$$S_{t,w,\alpha}(x_t) - S_{t+1,w,\alpha}(x_{t+1}) \leq \frac{2M(1 - \alpha^w)}{W(1 - \alpha)} \tag{7}$$

Proofs of the above lemmas are given in Sections A.1, A.2, A.3 in supplementary material.

**Theorem 3.4.** *Let the assumptions defined above are satisfied, $\eta = 1/\beta$, and $\alpha \to 1^-$, then Algorithm 2 guarantees an upper bound for the regret in Definition 2.3 as:*

$$DLR_w(T) \leq \frac{T}{W}\left(8\beta M + \sigma^2\right) \tag{8}$$

*which can be made sublinear in $T$ if $w$ is selected accordingly.*

Proof is given in section A.4 in supplementary material. This theorem justifies our use of a window and an exponential parameter $\alpha$ that approaches 1 from the left. One interesting observation is that Algorithm 2 is equivalent to momentum based SGD [Sutskever et al., 2013] when $T = w$. As a consequence, our contribution can be seen as a justification for the use of momentum in online learning by appropriate choice of regret.

## 4 Forecasting Overview

Standard mean squared error as a loss function summarizes the average relationship between inputs and outputs. The resulting forecast will be a point forecast which is the conditional mean of the value to be predicted given the input values, i.e. the most likely outcome. However, point forecasts provide only partial information about the conditional distribution of outputs. Many business applications such as inventory planning require richer information than just the point forecasts. Quantile loss, on the other hand, minimizes a sum that gives asymmetric penalties for overprediction and underprediction. For example, in demand forecasting, the penalty for overprediction and underprediction could be formulated as overage cost and opportunity cost, respectively. Hence, the loss for the ML model can be designed so that the profit is maximized. Therefore, using quantile loss as an objective function is often desirable in forecasting applications. The quantile loss for a given quantile $q$ between true value $y$ and the forecast value $\hat{y}$ is defined as:

$$L_q(y, \hat{y}) = q \max(y - \hat{y}, 0) + (1 - q) \max(\hat{y} - y, 0) \tag{9}$$

where $q \in (0, 1)$. Typically, forecasting systems produce outputs for multiple quantiles and horizons. The total quantile loss function to be minimized in such situations can be written as: $\sum_t \sum_k \sum_q L_q(y_{t+k}, \hat{y}^q_{t+k})$ where $\hat{y}^q_{t+k}$ is the output of the ML model, e.g. RNN, to forecast the q-th quantile of horizon k at forecast creation time t. This way, the model learns several quantiles of the conditional distribution such that $\mathbb{P}\left(y_{t+k} \leq y^q_{t+k} \mid y_{:t}\right) = q$. We use quantile loss as our cost function in the following section to forecast electric demand values from a time-series data set.

## 5 Experimental Results

We conduct experiments on a real-world time series dataset to evaluate the performance of our approach and compare with other SGD algorithms.

### 5.1 Time Series Data set

We use the data from GEFCom2014 [Barta et al., 2017] for our experiments. It is a public dataset released for a competition in 2014. The data contains 4 sub-datasets among which we use electrical loads. The electrical load directory contains 16 sub-directories: Task1-Task15 and Solution of Task 15. Each Task1-Task15 directory contains two CSV files: benchmark.csv and train.csv. Each train.csv file contains electrical load values per hour and temperature values measured by 25 stations. The train.csv file in Task 1 contains data from January 2005 to September 2010. The other folders have one month of data from October 2010 to December 2012. Each benchmark.csv file has benchmark forecasts of the electrical load values. These are point forecasts and score poorly on quantile loss metrics.

### 5.2 Implementation Details

The general flow chart of our experiments is illustrated in Figure 1(a). We use the data from January 2005 to September 2010 for training and we set the forecast time between October 2010 and December 2012. We assume that 5-year data arrives in monthly intervals. Therefore, we update the LSTM model every time new monthly data is observed. Computational details are given in Section A.5 in supplementary material.

**LSTM Model:** LSTMs are special kind of RNNs that are developed to deal with exploding and vanishing gradient problems by introducing input, output and forget gates [Hochreiter and Schmidhuber, 1997]. Our model contains two LSTM layers and three fully connected linear layers where each represents one of the three quantiles. The architecture of our LSTM model is illustrated in Figure 1(b). We use multi-step LSTM to forecast multiple horizons. We use electrial load value, hours of the

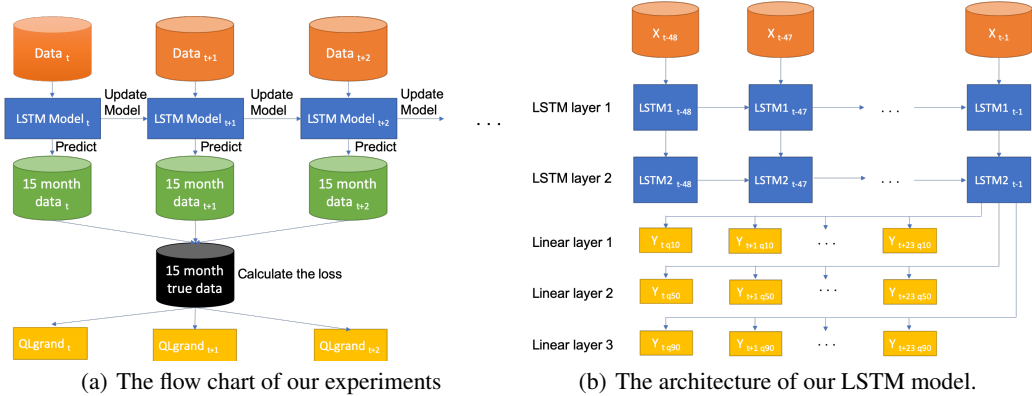

(a) The flow chart of our experiments   (b) The architecture of our LSTM model.

Figure 1: (a) Each data block in orange represents a month of data from the 5-year dataset. The model is updated each time a new month of data arrives. Our test set is the last 15 months of the dataset. Green blocks represent the forecasts for this period after each update. $QL_{grand}$ is computed using these forecasts and the true values in black. (b) We use multi-step LSTM to forecast multiple horizons. The input is two-day data of size $48 \times 44$ and the output is the prediction of three quantiles of next one-day electrical load values.

day, days of the week and months of the year as features so that the total number of features is 44. The input to our LSTM model is $48 \times 44$ where 48 is hours in two days. The output is the prediction of three quantiles of next day's values.

**Training:** During the update, we allow only one pass to the data, which means that the epoch number is set to 1. In order to make learning curves smoother, we adjust the learning rate at each update $t$ so that $\eta_t \leftarrow \eta/\sqrt{t}$ where $\eta$ is the initial value for the learning rate. In our experiments, we use 1, 3, 5, 9 for the value of $\eta$.

**Metrics:** After updating the model once, we evaluate the performance on the 15 months of test data (October 2010 - December 2012). We compute quantile loss for each month and report the average of these which we call $QL_{grand}$. Lower $QL_{grand}$ indicates better performance.

**Methods**: We use one offline and three online methods for training. The offline model uses the standard SGD algorithm and is re-trained from scratch on all data each time new data arrives. We see this strategy as the best strategy to be achieved, but as the most expensive in terms of computation. We call this SGD offline in our experiments. The online models are updated on new data as it is observed, without reviewing old data. We use standard SGD (called SGD online), static time-smoothed SGD (called STS-SGD) and our proposed dynamic exponentially time-smoothed SGD (called DTS-SGD) for online models.

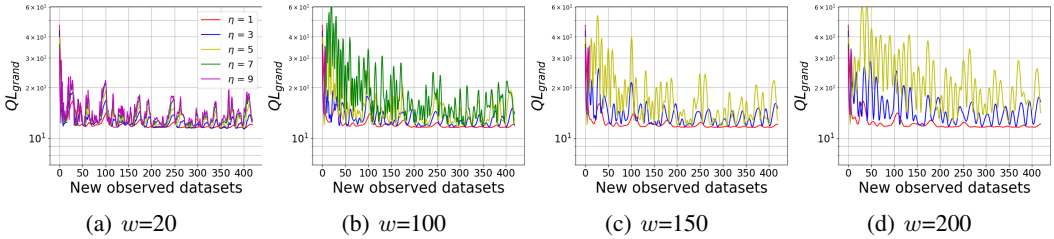

(a) $w$=20   (b) $w$=100   (c) $w$=150   (d) $w$=200

Figure 2: STS-SGD for different window sizes and learning rates. The learning curves become more sensitive to the selection of learning rates as the window size increases.

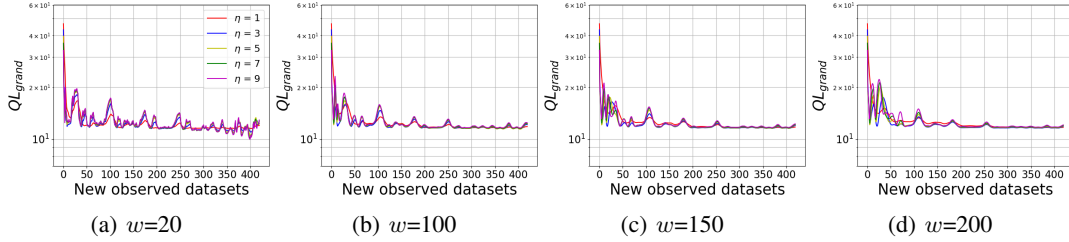

| (a) $w$=20 | (b) $w$=100 | (c) $w$=150 | (d) $w$=200 |

Figure 3: DTS-SGD (ours) for different window sizes and learning rates. The learning curves stay stable against different window sizes and learning rates.

## 5.3 Results

We compare the performance of online models in terms of their (i) accuracy, (ii) stability against window size, (iii) stability against the selection of learning rate, and (iv) computational efficiency.

**Stability Against Window Size:** Figures 2 and 3 show stability against window size for STS-SGD and DTS-SGD for different learning rates. As the window size increases, STS-SGD becomes more sensitive to the learning rate. The smoothest results with STS-SGD are obtained when the learning rate and the window size are small. For DTS-SGD, it takes longer for the curves to converge as the window size increases. However, it stays more stable against different learning rates regardless of window size.

**Stability Against Learning Rate:** We plot cumulative loss across $t$ as a function of learning rates in Figure 4 to evaluate sensitivity of three online learning methods to learning rates. It can be seen that DTS-SGD performs well for a wider range than STS-SGD and SGD online. STS-SGD started yielding *nan* (not a number) results due to very large value of losses as $\eta$ become larger; hence not shown in the figure. The minimum values of cumulative $QL_{grand}$ for each online method are: $14,612$ for SGD online, $14,585$ for DTS-SGD and $14,595$ for STS-SGD indicating that global minimums are very similar but DTS-SGD is marginally better. However, other approaches require more careful selection of a learing rate. SGD offline is not shown in this figure because it was computationally infeasible to compute SGD offline for such a wide range of learning rates. In Figure 5, we compare three online methods and SGD offline for relatively smaller range of learning rates. Each sub-figure shows performance as a

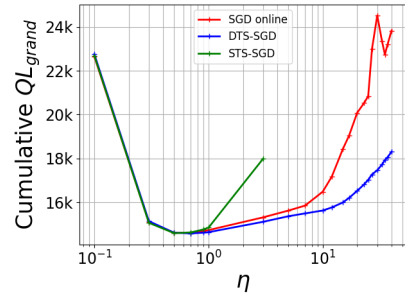

Figure 4: Comparison of online methods for their sensitivity to the learning rate. Our DTS-SGD performs well for a wider range of learning rates.

function of $t$ given a learning rate. The results show that larger learning rate is needed for SGD offline and it is the best performing model as expected. However, the results for SGD online and STS-SGD oscillate a lot indicating that they are very sensitive to the changes in learning rate as also observed in Figure 4. Our proposed approach DTS-SGD, on the other hand, stays robust as we increase the learning rate. Note that, for $\eta = 9$, the values for STS-SGD became *nan* (not a number) due to very large losses after some number of iterations, hence are not shown in the Figure.

We also ran experiments using SGD with momentum for various decay parameters and concluded that SGD with momentum is not even as stable as SGD-online (standard SGD without momentum) to large values of learning rate as shown in Figure A.1.

**Computation Time:** We further investigate the computation time of each method. Figure 6 shows the amount of time spent in terms of GPU seconds at each update for $\eta = 9$ and varying $w$ for STS-SGD and DTS-SGD. Note that, these results will not be different for other learning rates since computation time does not depend on the learning rate. The figure shows that the elapsed time increases for STS-SGD and DTS-SGD as $w$ increases as expected. It can be seen that the time elapsed curve looks exponential for SGD offline and linear for STS-SGD and DTS-SGD. As $w$ increases, both STS-SGD and DTS_SGD become slower but DTS-SGD is still more efficient that SGD offline.

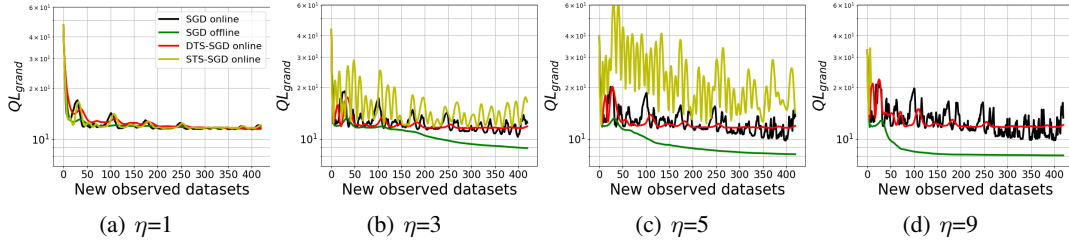

(a) $\eta$=1      (b) $\eta$=3      (c) $\eta$=5      (d) $\eta$=9

Figure 5: Comparison of models in terms of accuracy for various learning rates. Our DTS-SGD is less sensitive to $\eta$ than SGD online and STS-SGD. SGD offline performs the best as expected and yields higher accuracy as $\eta$ increases. Note that the values for STS-SGD become *nan* (not a number) after a few interations for $\eta = 9$ because of large values of gradients.

The reason why STS-SGD is not as efficient as DTS-SGD is because it needs to store previous losses and compute the gradients using the current parameters resulting in more backpropagation steps. Unsurprisingly, SGD online is the most efficient but its accuracy results in Figure 5 were not as stable as that of DTS-SGD.

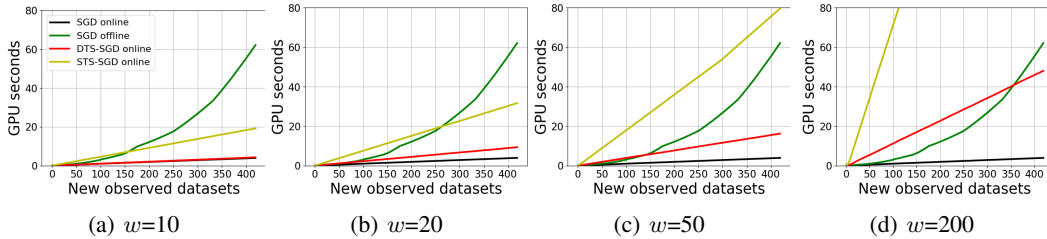

(a) $w$=10      (b) $w$=20      (c) $w$=50      (d) $w$=200

Figure 6: Comparison of computation time between four models with varying $w$ when $\eta = 9$. Computation time for STS-SGD and DTS-SGD increases as $w$ increases. Our DTS-SGD is more efficient than the SGD offline even for large $w$.

## 6    Conclusion

In this work, we introduce a local regret for online forcasting with non-convex models and propose dynamic exponentially time-smoothed gradient descent as an update rule. Our contribution is inspired by adapting the approach of Hazan et al. [2017] to forecasting applications. The main idea is to smooth the gradients in time when an update is performed using the new data set. We evaluate the performance of this approach compared to: static time-smoothed update, a standard online SGD update, and an expensive offline model re-trained on all past data at every time step. We use a real-world data set to compare all models in terms of computation time and stability against learning rate tuning. Our results show that our proposed algorithm DTS-SGD: (i) achieves the best loss on the test set (likely a statistical tie); (ii) is not sensitive to the learning rate, and (iii) is computationally efficient compared to the alternatives. We believe that our contribution can have a significant impact on applications for online forecasting problems.

## Acknowledgements

This project has been supported by AWS Machine Learning Research Awards.

## Footnotes

*www.sergulaydore.com

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
