[Supplementary Material · supp_mat.pdf]

# Supplementary Material for "Dynamic Local Regret for Non-convex Online Forecasting"

## A.1 Proof of Lemma 3.1

*Proof.* Due to the $\beta$-smoothness of $f_t$ functions, $S_t$ is $\beta$-smooth as well. Hence, we have:

$$
\begin{aligned}
S_{t,w,\alpha}(x_{t+1}) - S_{t,w,\alpha}(x_t) &\leq \langle \nabla S_{t,w,\alpha}(x_t), x_{t+1} - x_t \rangle + \frac{\beta}{2}\|x_{t+1} - x_t\|^2 \\
&= -\eta \left\langle \nabla S_{t,w,\alpha}(x_t), \tilde{\nabla} S_{t,w,\alpha}(x_t) \right\rangle + \eta^2 \frac{\beta}{2}\|\tilde{\nabla} S_{t,w,\alpha}(x_t)\|^2 - \eta\|\nabla S_{t,w,\alpha}(x_t)\|^2 \\
&\quad - \eta \left\langle \nabla S_{t,w,\alpha}(x_t), \tilde{\nabla} S_{t,w,\alpha}(x_t) - \nabla S_{t,w,\alpha}(x_t) \right\rangle + \eta^2 \frac{\beta}{2}\left(\|\nabla S_{t,w,\alpha}(x_t)\|^2\right) \\
&\quad + \eta^2 \frac{\beta}{2}\left(2\left\langle \nabla S_{t,w,\alpha}(x_t), \tilde{\nabla} S_{t,w,\alpha}(x_t) - \nabla S_{t,w,\alpha}(x_t) \right\rangle\right) \\
&\quad + \eta^2 \frac{\beta}{2}\left(\|\tilde{\nabla} S_{t,w,\alpha}(x_t) - \nabla S_{t,w,\alpha}(x_t)\|^2\right) \qquad (1) \\
&= -\left(\eta - \frac{\beta}{2}\eta^2\right)\|\nabla S_{t,w,\alpha}(x_t)\|^2 \\
&\quad - (\eta - \beta\eta^2)\left\langle \nabla S_{t,w,\alpha}(x_t), \tilde{\nabla} S_{t,w,\alpha}(x_t) - \nabla S_{t,w,\alpha}(x_t) \right\rangle \\
&\quad + \eta^2 \frac{\beta}{2}\|\tilde{\nabla} S_{t,w,\alpha}(x_t) - \nabla S_{t,w,\alpha}(x_t)\|^2. \qquad (2)
\end{aligned}
$$

Now, by applying $\mathbb{E}[. \mid x_t]$ on both sides of the above equation and using the result in equation 4, we prove the lemma:

$$
\begin{aligned}
\left(\eta - \frac{\beta}{2}\eta^2\right)\|\nabla S_{t,w,\alpha}(x_t)\|^2 &\leq \mathbb{E}\left[S_{t,w,\alpha}(x_t) - S_{t,w,\alpha}(x_{t+1})\right] + \eta^2 \frac{\beta}{2} \frac{\sigma^2(1 - \alpha^{2w})}{W^2(1 - \alpha^2)} \\
&= S_{t,w,\alpha}(x_t) - S_{t+1,w,\alpha}(x_{t+1}) + S_{t+1,w,\alpha}(x_{t+1}) - S_{t,w,\alpha}(x_{t+1}) \\
&\quad + \eta^2 \frac{\beta}{2} \frac{\sigma^2(1 - \alpha^{2w})}{W^2(1 - \alpha^2)}. \qquad (3)
\end{aligned}
$$

$\square$

## A.2 Proof of Lemma 3.2

*Proof.*

$$
\begin{aligned}
S_{t+1,w,\alpha}(x_{t+1}) - S_{t,w,\alpha}(x_{t+1}) &= \frac{1}{W} \sum_{i=0}^{w-1} \alpha^i \left( f_{t+1-i}(x_{t+1-i}) - f_{t-i}(x_{t+1-i}) \right) \\
&= \frac{1}{W} \left\{ f_{t+1}(x_{t+1}) - f_t(x_{t+1}) + \alpha f_t(x_t) - \alpha f_{t-1}(x_t) + \cdots \right. \\
&\qquad \left. + \alpha^{w-1} f_{t-w+2}(x_{t-w+2}) - \alpha^{w-1} f_{t-w+1}(x_{t-w+2}) \right\} \\
&= \frac{1}{W} f_{t+1}(x_{t+1}) - \frac{\alpha^{w-1}}{W} f_{t-w+1}(x_{t-w+2}) \\
&\quad + \frac{1}{W} \sum_{i=1}^{w-1} \alpha^i f_{t-i+1}(x_{t-i+1}) - \alpha^{i-1} f_{t-i+1}(x_{t-i+2}) \quad (4)\\
&\leq \frac{M\left(1+\alpha^{w-1}\right)}{W} + \frac{M(1-\alpha^{w-1})(1+\alpha)}{W(1-\alpha)} \quad (5)
\end{aligned}
$$

where the following inequality follows from $\frac{1}{W} f_{t+1}(x_{t+1}) - \frac{\alpha^{w-1}}{W} f_{t-w+1}(x_{t-w+2}) \leq \frac{M\left(1+\alpha^{w-1}\right)}{W}$ and $\frac{1}{W} \sum_{i=1}^{w-1} \alpha^i f_{t-i+1}(x_{t-i+1}) - \alpha^{i-1} f_{t-i+1}(x_{t-i+2}) \leq +\frac{M(1-\alpha^{w-1})(1+\alpha)}{W(1-\alpha)}$. $\qquad\square$

## A.3 Proof of Lemma 3.3

*Proof.* The proof simply follows from the boundedness property of $f_t$:

$$
\begin{aligned}
S_{t,w,\alpha}(x_t) - S_{t+1,w,\alpha}(x_{t+1}) &= \frac{1}{W} \sum_{i=0}^{w-1} \alpha^i \left( f_{t-i}(x_{t-i}) - f_{t+1-i}(x_{t+1-i}) \right) \\
&\leq \frac{2M(1-\alpha^w)}{W(1-\alpha)}. \quad (6)
\end{aligned}
$$

$\qquad\square$

## A.4 Proof of Theorem 3.4

*Proof.* Using the results from lemmas 3.1, 3.2 and 3.3, we can write the following inequality for $\|\nabla S_{t,w,\alpha}(x_t)\|^2$ as:

$$
\|\nabla S_{t,w,\alpha}(x_t)\|^2 \leq \frac{\frac{2M(1-\alpha^w)}{W(1-\alpha)} + \frac{M(1+\alpha^{w-1})}{W} + \frac{M(1-\alpha^{w-1})(1+\alpha)}{W(1-\alpha)} + \eta^2 \frac{\beta}{2} \frac{\sigma^2(1-\alpha^{2w})}{W^2(1-\alpha^2)}}{\eta - \frac{\eta^2 \beta}{2}} \quad (7)
$$

Substituting $\eta = 1/\beta$ yields:

$$\|\nabla S_{t,w,\alpha}(x_t)\|^2 \leq \frac{2\beta M}{W}\left(\frac{2(1-\alpha^w)}{1-\alpha} + (1+\alpha^{w-1}) + \frac{(1-\alpha^{w-1})(1+\alpha)}{(1-\alpha)}\right) + \frac{\sigma^2(1-\alpha^{2w})}{W^2(1-\alpha^2)}$$

$$\leq \frac{2\beta M}{W}\left(\frac{2(1-\alpha^w)}{1-\alpha} + (1+\alpha^{w-1}) + \frac{(1-\alpha^w)(1+\alpha)}{(1-\alpha)}\right) + \frac{\sigma^2(1-\alpha^{2w})}{W^2(1-\alpha^2)}$$

$$= \frac{2\beta M}{W}\left(\frac{1-\alpha^w}{1-\alpha}(3+\alpha) + (1+\alpha^{w-1})\right) + \frac{\sigma^2(1-\alpha^{2w})}{W^2(1-\alpha^2)}$$

$$\leq \frac{2\beta M}{W}\left(4\frac{1-\alpha^w}{1-\alpha} + (1+\alpha^{w-1})\right) + \frac{\sigma^2(1-\alpha^{2w})}{W^2(1-\alpha^2)}$$

$$\leq \frac{2\beta M}{W}\left(4\frac{1-\alpha^w}{1-\alpha} + \frac{(1+\alpha^{w-1})}{1-\alpha}\right) + \frac{\sigma^2(1-\alpha^{2w})}{W^2(1-\alpha^2)}$$

$$\leq \frac{8\beta M}{W}\left(\frac{1-\alpha^w}{1-\alpha} + \frac{(1+\alpha^{w-1})}{1-\alpha}\right) + \frac{\sigma^2(1-\alpha^{2w})}{W^2(1-\alpha^2)}$$

$$= \frac{8\beta M}{W}\left(\frac{2-\alpha^w+\alpha^{w-1}}{1-\alpha}\right) + \frac{\sigma^2(1-\alpha^{2w})}{W^2(1-\alpha^2)} \tag{8}$$

As $\alpha \to 1^-$, we get the following inequality:

$$\lim_{\alpha \to 1^-}\|\nabla S_{t,w,\alpha}(x_t)\|^2 \leq \frac{1}{W}\left(8\beta M + \sigma^2\right) \tag{9}$$

Summing the above inequality over $T$ concludes the proof. $\qquad\square$

## A.5   Computational Details

We use Python 3.7 for implementation [Oliphant, 2007] using open source library PyTorch [Paszke et al., 2017]. We use 2 NVIDIA GeForce RTX 2080 Ti GPUs with 512 GB Memory to run our experiments.

## A.6   Comparison with Online SGD with Momentum

We compare our approach with SGD online with momentum. Figure A.1 shows that SGD online with momentum is not as robust as our DTS-SGD to large values of learning rate.

Figure A.1: SGD online with momentum