[Reviews · NeurIPS 2019]

Reviewer 1



- This paper considers the online forecasting problem for non-convex machine learning methods. It first introduces a new concept to measure the regret of the online learning algorithm. It then proposes a method with sublinear regret (under the new regret definition). Experiments are not very comprehensive, but sufficient for a theoretically inclined paper. - The problem being studied by this paper is important and interesting: non-convex machine learning with gradient-based optimization. The proposed concept for dynamic regret is simple and intuitive. The theorems and results are clearly stated and proved. - The proposed technique of using exponential weights is sound, but seems rather straight forward and incremental compared with Hazan et al.

Reviewer 2



This paper introduces a local regret for non-convex models in a dynamic environment. The authors present an update rule incurring a cost that is sublinear in time T. It evaluates its performance on a real world dataset and shows improved performance compared with baselines. I am outside of the area and could not comment on the significance of the theoretical results. The experiments look reasonable to me.

Reviewer 3



This paper considers online forecasting problems with non-convex models. One of the main challenges in forecasting is concept drift, which refers to changes in the underlying relationship between inputs and outputs over time. Classic online learning algorithms have performance bounds in terms of regret, which compares the algorithm's performance to the best fixed action in hindsight. While regret makes sense when losses are convex, it is no longer appealing when there are non-convex losses. Hazan et al. 2017 introduced a notion of local regret to be used for online non-convex problems, as well as algorithms that achieve sublinear local regret. The local regret is a sum over time steps, where each term corresponds to the squared norm of the gradient of the average of the most recent loss functions. The main contribution of this paper is to propose a new form of local regret called dynamic local regret and to give algorithms for achieving low dynamic local regret. Dynamic local regret is similar to local regret in that it includes a sum of norms of gradients of some time-average. However, the time-average in dynamic local regret is weighted with some decay parameter and the functions within the time average are evaluated at previous inputs. The proposed algorithm for minimizing this regret is an exponentially time-smoothed SGD. The authors prove a bound on the dynamic local regret suffered by this algorithm. The authors also give some experimental results showing that their algorithm does well compared to the one from Hazan et al. 2017. On the positive side, this paper presents a new notion of regret for non-convex forecasting problems with concept drift. The authors also provide analysis and some theoretical bounds for their algorithm in terms of this new regret. The paper also gives a number of experimental results, showing that their algorithm is more stable and generally more computationally efficient than alternatives. On the negative side, the overall motivation and intuition behind the new definition of dynamic local regret isn't that compelling. Without some formal notion or even toy scenario for concept drift, it's not clear what theoretical basis there is to prefer this notion of regret to other notions other than some vague heuristics. Empirically, while the authors' algorithm (PTS-SGD) seems to be more stable and faster than the algorithm by Hazan et al. 2017, SGD seems to perform roughly as well as PTS-SGD, albeit with less stability. However, as the authors note, previous work has shown that PTS-SGD coincides with SGD with momentum as the decay factor approaches 1, so it seems like a better empirical comparison might be with SGD with momentum. Overall, this paper is a reject. The new notion of regret is interesting, but seems to need more theoretical justification to warrant why one would use this notion of regret instead of more established notions. The algorithm given is very similar to previous algorithms, and while the experiments show some interesting stability property of PTS-SGD, they do not give more evidence that dynamic local regret is important. The contributions seem like small modifications of previous results, and the benefits seem relatively unclear. The paper is reasonably clear, but could use more motivation or explanation of the notion of regret. ======== Post-rebuttal ======= I have read the other reviews and the rebuttal. I appreciated the toy example, the theoretical motivation via calibration, and the commends on SGD with momentum. I am happy with accepting the paper provided that the authors include the content from the rebuttal in the final paper.

[Author Response · NeurIPS 2019]

1. **REVIEWER 2** Thank you for your encouraging comments.

2. **REVIEWER 3** Thank you for your comments. In order to help clarify our contributions and or-
3. ganize them for readers, we provide the following table to summarize the differences between regrets.

| Regret | Non-convex Models | Concept Drift | Update Rule |
|---|---|---|---|
| Standard Regret | ✗ | ✗ | $x_{t+1} = x_t - \frac{\eta}{\sqrt{t}} \hat{\nabla} f_t(x_t)$ |
| Static Local Regret (Hazan et al.) | ✓ | ✗ | $x_{t+1} = x_t - \frac{\eta}{w} \sum_{i=0}^{w-1} \hat{\nabla} f_{t-i}(x_t)$ |
| Dynamic Local Regret (Ours) | ✓ | ✓ | $x_{t+1} = x_t - \frac{\eta}{W} \sum_{i=0}^{w-1} \alpha^i \hat{\nabla} f_{t-i}(x_{t-i})$ |

5. **REVIEWER 4** Thank you for your comments. First we provide a toy example and some additional theoretical
6. motivation for our regret in response to the following comment:

7. <span style="color:red">Without some formal notion or even toy scenario for concept drift, it's not clear what theoretical basis there is to prefer</span>
8. <span style="color:red">this notion of regret to other notions other than some vague heuristics.</span>

9. **Motivation via a Toy Example** We demonstrate the motivation of our dynamic regret via a toy example where the
10. static local regret fails. Concept drift occurs when the optimal model at time $t$ may no longer be the optimal model
11. at time $t + 1$. Consider an online learning problem with concept drift with $T = 3$ time periods and loss functions:
12. $f_1(x) = (x-1)^2, f_2(x) = (x-2)^2, f_3(x) = (x-3)^2$. Obviously, the best possible sequence of parameters is
13. $x_1 = 1, x_2 = 2, x_3 = 3$. Call this the *oracle policy*. Also consider a suboptimal sequence, where the model does not
14. react quickly enough to concept drift: $x_1 = 1, x_2 = 1.5, x_3 = 2$. Call this the *stale policy*. The values of the *stale*
15. *policy* were chosen to minimize Static Local Regret. Recall the formulation of static and dynamic local regrets:

$$HR_3(3) = \left\| \frac{\nabla f_3(x_3) + \nabla f_2(x_3) + \nabla f_1(x_3)}{3} \right\|^2 + \left\| \frac{\nabla f_2(x_2) + \nabla f_1(x_2)}{3} \right\|^2 + \left\| \frac{\nabla f_1(x_1)}{3} \right\|^2 \quad \text{(Hazan's)}$$

$$PR_3(3) = \left\| \frac{\nabla f_3(x_3) + \nabla f_2(x_2) + \nabla f_1(x_1)}{3} \right\|^2 + \left\| \frac{\nabla f_2(x_2) + \nabla f_1(x_1)}{3} \right\|^2 + \left\| \frac{\nabla f_1(x_1)}{3} \right\|^2 \quad \text{(Ours)}$$

17. Note that, for the local regrets, we use $w = 3$ and assume $f_t(x) = 0$ for $t \leq 0$. We also set $\alpha = 1$ for our Dynamic
18. Local Regret but other values would not change the results for this example. The formulation of the Standard Regret
19. is $\sum_{t=1}^{T} f_t(x_t) - \min_x \sum_{t=1}^{T} f_t(x)$. Although the *oracle policy* achieves globally minimal loss, Hazan et al.'s Static Local
20. Regret favors the *stale policy*. We can verify this by computing the loss and regret for these policies, as shown in the
21. table below.

| Regret | Oracle Policy | Stale Policy | Decision |
|---|---|---|---|
| Cumulative Loss | 0 | 5/4 | Oracle policy is better |
| Standard Regret | -2 | -3/8 | Oracle policy is better |
| Static Local Regret (Hazan et al.) | 40/9 | 4/9 | <span style="color:red">Stale policy is better</span> |
| Dynamic Local Regret (Ours) | 0 | 10/9 | Oracle policy is better |

Figure 1: SGD online with momentum

23. **Theoretical motivation via Calibration:** A more formal motivation of our regret
24. can be related to the concept of calibration [1]. The comment on line 110 can be
25. rewritten as: *If the updates $\{x_1, \cdots, x_T\}$ are **well-calibrated**, then perturbing $x_t$ by*
26. *any $u$ cannot substantially reduce the cumulative loss.* Hence, it can be said that the
27. sequence $\{x_1, \cdots, x_T\}$ is asymptotically calibrated with respect to $\{f_1, \cdots, f_T\}$ if:
28. $\limsup_{T \to \infty} \sup_{\|u\|=1} \frac{\sum_{t=1}^{T} f_t(x_t) - \sum_{t=1}^{T} f_t(x_t+u)}{T} \leq 0$. Consequently, using the first
29. order Taylor series expansion, we can write the following equation that motivates the left hand side of the equation 3 in
30. the paper: $\limsup_{T \to \infty} \sup_{\|u\|=1} -\frac{1}{T} \langle u, \nabla f_t(x_t) \rangle \leq 0$. Thus our regret ensures asymptotic calibration.

31. This analysis was dropped for simplicity, but thanks to the reviewer's comments we will put this analysis back into the
32. paper.

33. Next we provide some additional discussion of momentum to address the following comment:

34. <span style="color:red">However, as the authors note, previous work has shown that PTS-SGD coincides with SGD with momentum as the</span>
35. <span style="color:red">decay factor approaches 1, so it seems like a better empirical comparison might be with SGD with momentum.</span>

36. We indeed ran experiments using SGD with momentum for various decay parameters and concluded that SGD with
37. momentum is not even as stable as SGD-online (standard SGD without momentum) as shown in Figure 1. Our
38. PTS-SGD is still more robust to the learning rate. On the other hand, we observed that SGD with momentum yields
39. better accuracy for offline learning (results are not shown here). We will add these results to the paper.

40. # References

41. [1] Dean P Foster and Rakesh V Vohra. Asymptotic calibration. *Biometrika*, 85(2):379–390, 1998.


[Meta-Review · NeurIPS 2019]

The problem and formal model are interesting and general. The proposed objective is a non-straightforward modification of an existing criterion. It is subsequently optimized by adaptation of existing methods. The clarification provided in the author rebuttal (including the toy example) were very useful in appreciating the paper, and swung the reviewing process to accept the paper. Please incorporate the information from the rebuttal in the camera-ready version.